Whole genome phylogeny for 21 Drosophila species using predicted 2b-RAD fragments

Seetharam Arun S. 1 aseetharam@purdue.edu
Stuart Gary W. 2
1 Bioinformatics Core, Purdue University , West Lafayette, IN , USA
2 Department of Biology, Indiana State University , Terre Haute, IN , USA
Crandall Keith
Electronic publication date: 2013 Dec 23
Publication date: 2013
Volume: 1
Electronic Location ID: e226
Received 2013 Sep 26; Accepted 2013 Nov 28
Copyright: © 2013 Seetharam et al.
Copyright year: 2013
Copyright holder: Seetharam et al.
License: This is an open access article distributed under the terms of the Creative Commons Attribution License, which permits unrestricted use, distribution, and reproduction in any medium, provided the original author and source are credited.
License URL: https://creativecommons.org/licenses/by/3.0/

Keywords: Type IIB restriction enzymes, Phylogenomics, Restriction-site associated DNA (RAD) tags, Reduced genomic representation

Funding: Graduate Student Assistantship from the Biology Department and School of Graduate Studies, Indiana State University This work was supported by the Graduate Student Assistantship from the Biology Department and School of Graduate Studies, Indiana State University, held by Arun Seetharam. The funders had no role in study design, data collection and analysis, decision to publish, or preparation of the manuscript.

==============================
Type IIB restriction endonucleases are site-specific endonucleases that cut both strands of double-stranded DNA upstream and downstream of their recognition sequences. These restriction enzymes have recognition sequences that are generally interrupted and range from 5 to 7 bases long. They produce DNA fragments which are uniformly small, ranging from 21 to 33 base pairs in length (without cohesive ends). The fragments are generated from throughout the entire length of a genomic DNA providing an excellent fractional representation of the genome. In this study we simulated restriction enzyme digestions on 21 sequenced genomes of various Drosophila species using the predicted targets of 16 Type IIB restriction enzymes to effectively produce a large and arbitrary selection of loci from these genomes. The fragments were then used to compare organisms and to calculate the distance between genomes in pair-wise combination by counting the number of shared fragments between the two genomes. Phylogenetic trees were then generated for each enzyme using this distance measure and the consensus was calculated. The consensus tree obtained agrees well with the currently accepted tree for the Drosophila species. We conclude that multi-locus sub-genomic representation combined with next generation sequencing, especially for individuals and species without previous genome characterization, can accelerate studies of comparative genomics and the building of accurate phylogenetic trees.

Introduction

Evolutionary relationships of species derived by comparing single orthologous genes or groups of genes can be negatively affected by potential horizontal gene transfers, incomplete lineage-sorting, introgression, and the unrecognized comparison of paralogous genes (Delsuc, Brinkmann & Philippe, 2005). However, with the advent of the genomic era, it is now possible for researchers to use the complete genomes of fully sequenced organisms for building trees. Though such trees offer robustness for analysis, it becomes impractical to use traditional methods for constructing large scale alignments and for generating trees from these alignments, mainly because of their large size and their highly heterogeneous nature. As a result, there are now sophisticated methods that don’t rely on alignment and are optimized for large scale data. These methods generally use vector representation of genes (Qi, Luo & Hao, 2004; Stuart, Moffett & Leader, 2002) or features such as gene content (Huson & Steel, 2004; Snel, Bork & Huynen, 1999; Tekaia, Lazcano & Dujon, 1999), gene order (Bourque & Pevzner, 2002; Korbel et al., 2002), intron positions (Roy & Gilbert, 2005), or protein domain structure (Lin & Gerstein, 2000; Yang, Doolittle & Bourne, 2005).

Despite a strong recent interest in the various large-scale non-alignment methods, they are often viewed as somewhat less rigorous and less reliable. In addition, even with the dramatic decrease in the cost of genome sequencing, it is still not attractive to sequence the genomes of those organisms that have little economical value, especially if their genomes are extremely large. On the other hand, the possibility of obtaining a large and representative set of fragments, instead of the whole genome sequence, can be economically feasible even for the lesser known species and can provide a valuable alternative for many types of genomic scale studies, including phylogenomics.

Recently, several approaches have been developed to represent the genome by randomly sampling the entire genome. These approaches give a good reduced representation of the genome and are based on restriction sites on the genome combined with the next generation sequencing methods. Some popular methods include Complexity Reduction of Polymorphic Sequences (CRoPS) (van Orsouw et al., 2007); restriction site-associated DNA sequencing (RAD-seq) (Baird et al., 2008; Etter et al., 2011); Genotyping by Sequencing method (GBS); double-digest RAD-seq (Peterson et al., 2012), and 2bRAD (Wang et al., 2012). All these methods provides good subsamples from homologous locations within genomes and are widely used to study population genetics (Baxter et al., 2011; Hohenlohe et al., 2010). These methods have the potential to uncover detailed information about a wealth of genomic markers. Complex interactions among markers can also be extracted at the population level (Baird et al., 2008; Davey & Blaxter, 2010). Recently, these fragments have also been used for evolutionary studies (Emerson et al., 2010; Rubin, Ree & Moreau, 2012; Yi & Jin, 2013).

A novel class of enzymes, known as Type IIB restriction endonucleases (Roberts et al., 2003b), are site-specific endonucleases that cut both strands of double-stranded DNA upstream and downstream of their recognition sequences. These restriction enzymes have recognition sequences that are generally interrupted and range from 5 to 7 bases long. They produce DNA fragments which are of uniform length, ranging from 21 to 33 base pairs in length (without cohesive ends) (Roberts et al., 2003a). The fragments are generated from throughout the entire length of a genomic DNA providing an excellent fractional representation of the genome. This method of generating fragments using Type IIB enzymes is termed 2bRAD (Wang et al., 2012) and these fragments have been used for various purposes including population studies, digital karyotyping (Stebbins, 1950), for pathogen identification by computational subtraction (Tengs et al., 2004) and genomic profiling to identify and quantitatively analyze genomic DNAs (Dunn et al., 2002). In this study, we show that these fragments can be used for efficient phylogenetic study for determining evolutionary relationships between distinct species. We have tested this method in silico and shown that 13 different types of IIB restriction enzymes can be used to accurately reconstruct the phylogeny of a diverse set of 21 Drosophila species that are currently available.

Materials and Methods

Obtaining datasets

Whole genome, nucleotide sequences for the 21 Drosophila species were downloaded from the FlyBase (McQuilton, St Pierre & Thurmond, 2012), NCBI databases and from the Princeton University website (Rebeiz et al., 2009) on July 10, 2010.

Simulated restriction digestion

The PERL program “Phyper” was used to simulate restriction digestion for all 16 Type IIB endonuclease enzymes and for processing the obtained fragments. This program generated a representative list of unique fragments i.e., single-copy fragments (most abundant) and fragments that are present as multiple identical copies (less frequent). The remaining fragments belong to divergent fragment families within a given genome that display one or a few mutations relative to each other and were identified and removed from the analysis. The representative list of fragments were generated for each genome, for each enzyme separately.

Fragment comparisons

The representative lists of fragments were then used with another PERL program “Phyppa” for comparative analyses. This program compares each fragment of a genome with every fragment of another genome in order to find identical fragments and similar fragments (fragments with up to 5 mismatches for ensuring more than 80% similarity among sequences). A total of 210 such comparisons were done in order to generate the full list of shared fragments (identical fragments and similar fragments) for every pair of genomes (both PERL scripts are available upon request). Analyses was performed on a standard laptop with a quad core processor (1.73 GHz Intel Core i7) and with 6 GB RAM. For each enzyme, the scripts required about 6 h to finish for both fragment generation and comparison between all genomes.

Distance calculations

The number of shared fragments between a pair of genomes was then used to calculate the evolutionary distance by calculating the ratio of shared fragment to the total fragments and converting them to negative natural log (Eq. (1)). Conversion to negative natural log was essential to ensure that the distances computed were always positive. (1) Distance=−lnIdentical fragments+Similiar fragmentsTotal fragments of both species.

Building trees

Distance measures for all the pairwise comparisons for a particular enzyme were used to build trees using the neighbor program from the Phylip (Felsenstein, 2005) package. A consensus tree was them produced by combining trees for all the enzymes with the consensus program from Phylip. The flowchart for the entire process is given in Fig. 1.

Figure 1 Workflow of the entire process of generating phylogeny from the Type IIB fragments.

Results and Discussion

Datasets

The full nucleotide sequences for 21 Drosophila species downloaded from various sources are listed in Table 1. The genome size ranged from 137.82 mb for D. simulans to 235.52 mb for D. willistoni. D. willistoni had the lowest GC content of all with 37.89% and D. pseudoobscura had the highest GC content (45.43%).

Table 1 Various Drosophila species and source databases used for the analysis. The GC% for each genome was calculated using infoseq from the EMBOSS package.

Genome	GC%	Size	Source	
D. ananassae	42.56	230.99 mb	FlyBase	
D. biarmipes	41.82	168.58 mb	NCBI	
D. bipectinata	41.62	166.39 mb	NCBI	
D. elegans	40.31	170.51 mb	NCBI	
D. erecta	42.65	152.71 mb	FlyBase	
D. eugracilis	40.90	156.31 mb	NCBI	
D. ficusphila	41.93	151.04 mb	NCBI	
D. grimshawi	38.84	200.46 mb	FlyBase	
D. kikkawai	41.38	163.57 mb	NCBI	
D. melanogaster	42.05	168.73 mb	FlyBase	
D. mojavensis	40.22	193.82 mb	FlyBase	
D. persimilis	45.29	188.37 mb	FlyBase	
D. pseudoobscura	45.43	152.73 mb	FlyBase	
D. rhopaloa	40.07	193.90 mb	NCBI	
D. santomea	38.52	165.75 mb	Princeton University	
D. sechellia	42.53	166.57 mb	FlyBase	
D. simulans	43.06	137.82 mb	FlyBase	
D. takahashii	40.01	181.00 mb	NCBI	
D. virrilis	40.80	206.02 mb	FlyBase	
D. willistoni	37.89	235.51 mb	FlyBase	
D. yakuba	42.43	165.69 mb	FlyBase	

Type IIB restriction enzymes

The 16 Type IIB restriction endonucleases that could be used for simulating the restriction digestion of Drosophila genomes along with their recognition sites, average distance between the restriction sites assuming random distribution of nucleotides and without any compositional bias, and the size of fragment (blunt) that the enzymes leaves behind are given in Table 2 (Tengs et al., 2004). Unlike traditional Type II enzymes, Type IIB enzymes cleave on both sides of the recognition sequence (about 7–15 bases upstream and downstream, depending on enzyme) generating a fragment of uniform length. Also, the recognition site is usually split into two parts by some fixed number of random bases. They normally leave 2–3 base overhangs on the generated fragment.

Table 2 List of enzymes used for the fragment generation from the 21 Drosophila species.

Frequency indicates estimated distance between cut sites given a random sequence with all the 4 bases in equal probability and length refers to blunt tag length.

Enzyme	Recognition sequence	Frequency	Length	
AlfI	GCANNNNNNTGC	4096	32	
AloI	GAACNNNNNNTCC	8192	27	
BaeI	ACNNNNGTAYC	4096	28	
BcgI	CGANNNNNNTGC	2048	32	
BplI	GAGNNNNNCTC	4096	27	
BsaXI	ACNNNNNCTCC	2048	27	
BslFI	GGGAC	512	21	
Bsp24I	GACNNNNNNTGG	2048	27	
CspCI	CAANNNNNGTGG	8192	33	
FalI	AAGNNNNNCTT	4096	27	
HaeIV	GAYNNNNNRTC	1024	27	
PpiI	GAACNNNNNCTC	8192	27	
PsrI	GAACNNNNNNTAC	8192	27	

Fragment analyses

The numbers of representative fragments obtained from each genome for each enzyme are listed in Table 3. The most frequent cutting enzymes such as BslFI had generally higher numbers of fragments within all genomes compared to other enzymes. Also, D. pseudoobscura and D. persimilis had relatively higher numbers of fragments compared to other genomes with most of the enzymes. Following fragment extraction, the original genomic sequences downloaded from various source databases were represented as a collection of fragments of uniform length. For each genome a total of 16 fragment sets were generated by using 16 different type IIB enzymes. The number of fragments generated by each genome was not closely related to the size of their genomes but they were related to the GC content. Most of the enzymes used in the analysis recognized a GC rich recognition site which is reflected in the number of fragments generated with GC rich genomes. The genomes that were GC rich such as D. pseudoobscura and D. persimilis had higher numbers of fragments compared to other genomes. Similarly the genomes that had lower GC content such as D. willistoni and D. grimshawi generated fewer fragments. Overall, the number of fragments obtained for each species were within the range of expected fragments based on their genome size and estimated distance between restriction cut sites (assuming random sequence without GC content bias). Most enzymes predicted to be frequent cutters generated large number of fragments like BslFI. Predicted rare cutters like PsrI, PpiI, AloI and CspCI generated fewer fragments than other enzymes.

Table 3 Total number of fragments generated using 13 different Type IIB restriction enzymes for each of the 21 Drosophila genomes.

Genomes	AlfI	AloI	BaeI	BcgI	BplI	BsaXI	BslFI	Bsp24I	CspCI	FalI	HaeIV	PpiI	PsrI	
D. ananassae	34804	11421	6151	51646	21457	52433	101183	46042	16405	38109	74174	11193	8344	
D. biarmipes	41242	12667	6875	63518	22752	51248	109404	44554	18178	41284	75291	12177	10210	
D. bipectinata	35642	10893	6616	51208	20363	50001	98937	45563	17131	39286	73197	10545	8622	
D. elegans	43207	11314	6068	59905	18764	45496	93763	43259	18466	41866	75238	11027	9753	
D. erecta	42781	10517	5914	60434	18119	43684	85735	40020	17793	31931	66412	9979	8677	
D. eugracilis	36455	10170	5699	51988	18236	43177	86365	42020	17568	40795	72398	9682	8335	
D. ficusphila	38374	11698	5338	60448	20161	47056	89928	39223	17489	37380	69222	11070	8868	
D. grimshawi	49667	5891	5212	61420	17341	30379	58175	35658	16642	34409	64560	8062	6977	
D. kikkawai	39192	10361	5516	54698	21908	50258	99784	44066	16846	40965	68593	10765	8126	
D. melanogaster	39711	9908	6037	59203	16840	41168	81877	39221	17651	31350	68204	9243	8303	
D. mojavensis	54782	6294	5234	64186	21048	33289	60708	36674	14774	33071	65210	9090	8012	
D. persimilis	43327	10706	7567	59923	25287	53206	113002	48862	16329	31779	76473	12267	8940	
D. pseudoobscura	43650	10461	7466	60237	25174	53269	111423	48990	16358	31417	74808	12175	8774	
D. rhopaloa	36920	10920	6177	56203	18139	44894	93524	41357	17133	40153	76711	10442	9247	
D. santomea	40344	9877	5957	56771	17044	41850	80010	38107	17037	32142	67070	9414	8378	
D. sechellia	39876	10371	5808	59204	17430	42659	83936	39380	17276	31541	68359	9792	8289	
D. simulans	38549	9815	5547	56820	16777	40735	79826	37436	16666	30304	64321	9148	7773	
D. takahashii	37489	11463	5431	58887	19189	45240	91825	39992	26269	37277	74002	10801	8987	
D. virrilis	58785	6943	5774	64912	18097	31951	66710	38679	15733	37692	65275	9290	8551	
D. willistoni	34033	7083	6177	43299	15103	35578	70085	39996	17240	42202	77102	7941	9626	
D. yakuba	42202	10300	6165	59442	17885	43748	83095	39920	18007	33024	69632	9887	8765	

Distance matrices and phylogenetic trees

A comparison of fragments between genomes provided a list of fragments that were shared by those genomes. Closely related organisms are expected to share higher numbers of similar fragments (including identical fragments) compared to other distantly related genomes. Similar fragments are defined as those with 6 or fewer mismatches. Since the average length of fragments generated from various enzymes was around 27 bases, allowing 5 bases mismatch ensured at least 80% similarity among the sequences. The fragments being compared between 2 genomes ranged from 21 bp to 33 bp long (average size of 27 bp). The identical fragments between the 2 genomes are most likely to represent homologous or even orthologous sections of the genomes. Even for a fragment length of 21 bp (smallest fragment size produced by these enzymes), the probability that a particular 21 bp sequence exists one or more times in a genome of 150 Mb is 0.00341%. The pair-wise distance matrices constructed using the similar fragments detected by each enzyme were used to estimate phylogenetic trees (Fig. 2). The individual NJ trees obtained for each enzyme were largely consistent with the currently accepted relationships among the various Drosophila groups and subgroups, as was the single consensus tree obtained (Fig. 3). Per cent support values were calculated based on number of enzymes supporting the particular branch.

Figure 2 The consensus phylogenetic tree obtained by combining the trees obtained for each of the 13 enzymes.

The phylogenetic tree for each enzyme was calculated by extracting the corresponding fragments and then counting the number of shared fragment between every pair of species. The upper branch support values represent the percentage agreement over 13 enzymes and the bottom values indicate number of enzymes out of total 13 enzymes supporting the branch.

Figure 3 Single enzyme tree (AloI enzyme) showing the branch length.

Conclusions

The 21 species of Drosophila used here included the subgenus Sophophora and the subgenus Drosophila. The Sophophora group was represented by melanogaster, obscura and willistoni and the Drosophila group was represented by virilis, repleta and mojavensis. Out of the 12 subgroups within the melanogaster group, 9 subgroups viz., ananassae, montium, melanogaster, suzukii, takahashii, ficusphila, elegans, rhopaloa and eugracilis were represented by 15 species. Of these, only 2 subgroups had multiple members within our data set, but both displayed a monophyletic arrangement within the final tree shown in Fig. 2. The placement of the 12 well-studied Drosophila species viz., D. simulans, D. sechellia, D. melanaogaster, D. erecta, D. ananassae, D. yakuba, D. pseudoobscura, D. persimilis, D. willistoni, D. mojavensis, D. virilis and D. grimshawi within our tree corresponds exactly to the currently accepted phylogeny (Clark et al., 2007; Hahn, Han & Han, 2007; Haubold & Pfaffelhuber, 2012; Stark et al., 2007).

Overall, the topology of our 21 species tree agrees precisely with those presented by van der Linde et al. (2010), Haubold & Pfaffelhuber (2012) and Yang et al. (2012) and all the branches were completely resolved. The subgenus Sophophora was clearly distinguished into old world clades melanogaster/obscura and neo world clade willistoni in our tree (van der Linde & Houle, 2008). The largest group melanogaster, had multiple subgroups viz., melanogaster, montium, ananassae and oriental subgroup cluster (eugaracilis, suzukii, takahashii, elegans, rhopaloa, ficusphila). Many previous studies have failed to completely resolve the nodes within the oriental subgroup cluster (Da Lage et al., 2007; Toda, 1991). In our tree, ananassae group formed the earliest branch in the melanogaster group followed by montium subgroup with strong branch support values. Most of the earlier studies confirmed this topology (Da Lage et al., 2007; Kopp, 2006; Prud’homme et al., 2006) except for two studies that placed them together as a sister clade from the rest of the subgroups (Schawaroch, 2002) or reversed the order of branching (Yang et al., 2004). Both these studies had poor branch support. The oriental subgroups cluster formed three sub-clades. The first sub-clade included elegans and rhopaloa with ficusphila as the sister sub-group, the second sub-clade included suzukii and takahashii and the third sub-clade included the eugracilis sub-group. The placements of these sub-clades were controversial among the literature surveyed and was attributed to the explosive radiation of these oriental groups (van der Linde & Houle, 2008). The eugracilis clade consisting of D. eugracilis is most inconsistently placed clade and it is either placed as sister species of melanogaster sub group, as in our tree (Haubold & Pfaffelhuber, 2012; Pelandakis & Solignac, 1993; van der Linde et al., 2010) or as sister species of the sub clade formed by suzukii and takahashii (Yang et al., 2004) or as sister species of elegans and rhopaloa within the elegans — rhopaloa — ficusphila clade (Yang et al., 2012). The placements of the other two clades, suzukii — takahashii and elegans — rhopaloa — ficusphila within the melanogaster group in our tree is in agreement with other published studies (Kopp, 2006; Kopp & True, 2002). The sub-clade formed by suzukii and takahashii is well supported by most studies including ours with the strong branch support (Da Lage et al., 2007; Kopp & True, 2002; Schawaroch, 2002; Yang et al., 2004). Most studies have confirmed that the rhopaloa subgroup is the sister group of the elegans subgroup but the ficusphila sub group is considered to be polytomic branching clade in the melanogaster group (van der Linde & Houle, 2008). However, in our tree ficusphila sub group is presented as the sister species of rhopaloa — elegans subgroups, albeit with low branch support. Within the Drosophila subgenus, all three groups (virilis, repleta and grimshawi) exhibited a topology frequently observed in other studies (van der Linde & Houle, 2008).

A variety of sub-genomic sampling methods have been used previously for population studies and are especially effective on non-model organisms, but are rarely used for generating phylogenies for a diverse set of distinct species. We show here that multi-locus data obtained from short sub-genomic fragment sets, essentially 2b-RAD, provides good phylogenetic signal and produces a well resolved and well-supported species phylogeny. The wide adoption of various RAD-like methods is due to the fact that deep sequencing of the fragments produced can be easily accomplished following two simple steps: adapter ligation, and then PCR. These methods are applicable to any organism irrespective of its genome size. The 2b-RAD approach to fragment generation and characterization in particular is simple, quick and cost effective (Wang et al., 2012). This method also shares some similarity with the recently described, alignment free multi-locus “co-phylog” method (Yi & Jin, 2013). Both use a large number of short homologous fragments and, consequently, both can be profitably applied to short sequence reads derived via next generation sequencing, even prior to assembly. However, the co-phylog method is distinct in that it makes use of standard alignment algorithms applied to each locus to generate estimates of relatedness for building phylogenies. Effective application of the co-phylog method generally requires that the genomes being compared be closely related, and this would be expected to be true for our method as well, since effective matching of homologous short fragments in either case requires a significant degree of local sequence similarity. Despite this expected limitation, we note that the Drosophila species compared herein are relatively diverse, spanning approximately 40–50 million years of evolution.

Additional Information and Declarations

Competing Interests

Author Contributions

The authors declare they have no competing interests.

Arun S. Seetharam performed the experiments, analyzed the data, wrote the paper.

Gary W. Stuart conceived and designed the experiments, contributed reagents/materials/analysis tools, wrote the paper.

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
