# Peer review of "Whole genome phylogeny for 21 Drosophila species using predicted 2b-RAD fragments"

_PeerJ, doi:10.7717/peerj.226_

## Round 0.1 · original submission · Minor Revisions

I was able to get three reviews of your paper by area experts and they unanimously recommend publication with minor revisions. All have provided some minor comments to attend to below. Please address these and we'll be good. Thanks for submiting to PeerJ!

Reviewer 1 ·

Basic reporting

This article was, for the most part, well written and clear. The authors did a nice job of introducing the various "subgenomic" approaches that are being considered. I would have liked to see more discussion of the literature concerning the relationships of the species in the study, particularly within the subgenus Sophophora. Several key studies are not cited (Remsen and O'Grady 2002; O'Grady and Kidwell, 2002; Kopp and True, 2002. Minor edits: italics for all genera (Drosophila), species groups and subgroups.

Experimental design

The paper clearly fits with the scope of the journal and the methods are adequate. Since many phylogenetic studies use sequence data, rather than distance data, and much more sophisticated algorithms than Phylip, it would be nice to see a comparison of how the proposed distance method performs side-by-side with a Bayesian or ML analysis of the underlying sequences. More discussion of how the homology assessments were made is also necessary.

Validity of the findings

The phylogeny looks reasonable and is concordant with the tree you get using other methods with fewer data. The authors ran bootstrap analyses but do not talk about these results specifically in the text. This would be helpful in the discussion of the rhopaloa-elegans-ficusphila relationship, one of the disagreement points between previous work and the least well supported lineage in the current tree. I also found all the discussion of subclade of this versus subclade of that confusing. Why not just use the term clade and mention the different placement (along with support values)? Perhaps an inset showing some of the major alternative placements on your figure would be helpful for the discussion.

Reviewer 2 ·

Basic reporting

Seetharam & Stuart propose a method for phylogeny reconstruction from
whole genomes by sequencing restriction fragments of length 21--33
bp. They demonstrate the feasibility of their method by simulating its
application to 21 Drosophila genomes.

- The Title promises a phylogeny reconstruction method, but the paper
delivers the simulation of such a method. Please clarify.
- End of Abstract. "improve studies of comparative genomics and the
building of accurate phylogenetic trees.'' No improvement over
existing methods is shown.
- Introduction. Reference the co-phylog method mentioned in
the Discussion.
- End of first para. Haubold & Pfaffelhuber (2012) computed a tree of the same 21
Drosophila genomes using the alignment-free method of Domazet-Loso &
Haubold (2009).
- L. 41: "widely used to study population genetics''. References missing.

Experimental design

Section 2.2: Software mentioned here ought to be made available.
Section 2.2: "removing all closely related fragments''. How were
these defined and removed?

Validity of the findings

- The validity of the method ought to be demonstrated by an actual
restriction digest for one pair of the genomes investigated.
- Section 3.2: Why are there frequencies > 1 in Table 2? This table
ought to report "Expected Count'' conditional on GC content. For
AlfI & D. ana this is
0.43^4 * (1-0.43) * 231 * 10^6 = 4.5 * 10^6,
which is an order of magnitude larger than the actual result of
34,805. Please comment.
- Section 3.4: How are these matches with 6 mismatches computed?
- Section 3.4: What is the memory and time requirement of the
proposed distance computation method?
- Section 3.4: Annotate tree with bootstrap-like values that indicate
how many out of the 16 enzymes support a given clade.
- Section 3.4: Show tree for single enzyme with branch lengths.
- L. 170: Co-phylog is an alignment-free method.

·

Basic reporting

Interesting paper. The concept of using long sequences as characters in phylogenetics Is well known. Using REs to do phylogeny is also classic. Longer sequences have more character states so they are less prone to homoplasy. Hence there is soundness to thinking this approach would be precise.

The only problem I have with the paper is that it uses only a phenetic approach to tree building. With more attention to detail a character state matrix could also be constructed and parsimony and simple likelihood models could be applied. This would require that exact homology of fragments and sites be established. It would also result in more data. Sites would be characters not fragments. To have a fragment you need two sites. There is a lot of literature on this kind of analysis. Another advantage of treating things like characters is that Dollo parsimony can be used.

Experimental design

See above.

Validity of the findings

Highly valid if you like distance approaches.

Additional comments

Would it be possible in the light of thoroughness to get a character matrix and analyze that? You should have that info. If a site exists in an exon or in a five or three prime region you can establish homology fairly easily. Synteny for others would do the trick. Then analyze as characters.

---

## Round 0.2 · accepted · Accept

Thank you for effectively revising your manuscript to accommodate the reviewers' concerns. I feel the paper is now ready for publication.